# Identification of DDX60 as a Regulator of MHC-I Class Molecules in Colorectal Cancer

**DOI:** 10.3390/biomedicines10123092

**Published:** 2022-12-01

**Authors:** Nina Geng, Tuo Hu, Chunbo He

**Affiliations:** 1Eppley Institute for Research in Cancer and Allied Diseases, University of Nebraska Medical Center, Omaha, NE 68198, USA; 2Collège Jean-de-Brébeuf, Montréal, QC H3T 1C1, Canada; 3Department of Colorectal Surgery, The Sixth Affiliated Hospital of Sun Yat-sen University, Guangzhou 510655, China

**Keywords:** colorectal cancer, immunotherapy, MHC-I, DDX60

## Abstract

Immune checkpoint blockade (ICB) therapies induce durable responses in approximately 15% of colorectal cancer (CRC) patients who exhibit microsatellite instability-high (MSI-H) or deficient mismatch repair (dMMR). However, more than 80% of CRC patients do not respond to current immunotherapy. The main challenge with these patients is lack of MHC-I signaling to unmask their cancer cells so the immune cells can detect them. Here, we started by comparing IFNγ signature genes and MHC-I correlated gene lists to determine the potential candidates for MHC-I regulators. Then, the protein expression level of listed potential candidates in normal and cancer tissue was compared to select final candidates with enough disparity between the two types of tissues. ISG15 and DDX60 were further tested by wet-lab experiments. Overexpression of DDX60 upregulated the expression of MHC-I, while knockdown of DDX60 reduced the MHC-I expression in CRC cells. Moreover, DDX60 was downregulated in CRC tissues, and lower levels of DDX60 were associated with a poor prognosis. Our data showed that DDX60 could regulate MHC-I expression in CRC; thus, targeting DDX60 may improve the effects of immunotherapy in some patients.

## 1. Introduction

Colorectal cancer (CRC) is one of the most common cancers worldwide with 1.94 million new cases and 940,000 deaths in 2020 [1]. Although the 5-year survival rate of localized CRC is 90%, it drops to 15% if the cancer is diagnosed at advanced stages [2]. Surgery, chemotherapy, radiation, and targeted therapy are still the preferred treatments for CRC. Immunotherapies, including adoptive cell transfer and immune checkpoint blockade (ICB), have led to multiple breakthroughs across different cancer types in many clinical studies recently [3]. For instance, ICB, such as anti-PD-1, anti-PD-L1, and anti-CTLA-4, has been shown to induce remarkable responses in approximately 15% of CRC patients with microsatellite instability-high (MSI-H) and/or deficient mismatch repair (dMMR). However, most CRC patients do not respond to current immunotherapy [4]. There is an urgent need to find ways to increase the efficiency of ICB treatment in CRC.

The mechanisms of immunotherapy resistance are under extensive investigation, and several immune escape mechanisms have been identified, including low tumor infiltrating immune cells [5,6], potent immunosuppressive microenvironment [7,8,9], low tumor antigen load [10], defect of antigen presentation [11,12,13], loss of HLA expression [14,15,16], as well as other mechanisms [10,17,18,19]. Among these mechanisms, lacking antigen presentation to activate T cells by major histocompatibility complex (MHC-I) comprises a large proportion of patients. MHC-I molecules are expressed on the cell surface of all nucleated cells and present intracellular peptide fragments of proteins to cytotoxic T cells, allowing CD8 T cells to identify pathological cells synthesizing abnormal proteins, such as cancers expressing mutated proteins [20,21,22]. Studies demonstrated that downregulation of MHC-I-signaling pathways is a major mechanism supporting cancer immune evasion, correlating with a worse prognosis among patients. Since ICB-based treatment largely depends on the recognition of tumor cells by cytotoxic T cells, impaired MHC-I antigen processing and presentation often indicate resistance to ICB treatment [14,16]. Thus, restoring MHC-I expression might be an important way to improve current immunotherapy [23]. However, it is still largely unknown how MHC-I molecules are reduced in CRC cells.

The MHC-I complex consists of a constant heavy chain, encoded by the human leukocyte antigen (HLA)-A, HLA-B, and HLA-C genes, and an invariant light chain, encoded by a gene called β2-microglobulin (B2M) [24]. MHC-I components (HLA-A/B/C and B2M) are widely expressed in all nucleated cells and tightly regulated by intrinsic cellular stress (e.g., DNA damage) and extrinsic signals (e.g., stimulation of interferons) [25]. During viral infection, type I interferons (mainly secreted by infected cells) and type II interferon (interferon-gamma, IFNγ, mainly secreted by T cells) can induce the expression of the MHC-I component genes through the IFNγ-IFNGR-JAK-STAT pathway [26] and facilitate the CD8+ T-cell-dependent infected cell elimination [27]. In tumors, IFNγ secreted by activated tumor-infiltrating T cells can induce the expression of the MHC-I component genes in cancer cells [28], boosting the tumor antigen presentation and enhancing the CD8+ T-cell-mediated cancer cell clearance [28,29]. Both ex-vivo and in-vivo studies have established a connection between genetic and epigenetic alterations of the interferon and MHC-I signaling pathways and immunotherapy resistance. For example, tumors with copy number alterations in MHC-I and IFNγ-signaling showed adaptive resistance to ICB immunotherapy in patients [30,31]. Some mutations in JAK1/2, IFNGR1/2, and B2M may result in loss of MHC-I expression on the cell surface; these patients showed intrinsic resistance to immunotherapy [30,31,32,33]. However, genetic alterations of genes in IFN-γ and MHC-1 pathways are infrequently detected in most cancer patients. For example, none of the gene mutation rates of IFNGR1 (1.7%), IFNGR2 (1.5%), JAK1 (4%), JAK2 (4%), STAT1 (4%), HLA-A (3%), HLA-B (5%), HLA-C (3%), or B2M (6%) are found in excess of 6% in patients with CRC based on 526 samples data from TCGA datasets [34]. Other factors, such as the embryonic transcription factor double homeobox 4 (DUX4), lymphocyte adapter protein (LNK), the ubiquitin ligases RING finger protein 2 (RNF2), and the protein kinase D2 (PKD2) have been implicated in impaired IFN and MHC-I signaling in different cancers [14]. In addition, recent studies have shown that the MHC-I level can be also regulated at the post-transcriptional and post-translational levels [14]. For instance, upregulation of some microRNAs (miRNA) can inhibit the level of various parts of the MHC-I complex, including TAP1/TAP2 and calreticulin. After being translated, MHC-I molecules can be selectively targeted for lysosomal degradation through an autophagy-dependent mechanism in cancer cells. MHC-I molecules could be also stacked in the endoplasmic reticulum and be further subjected to ubiquitination and proteasomal degradation after their retro-translocation [35]. Thus, the MHC-I level could be restored at different levels to improve anti-tumor immune response.

In this study, we focused on the integrity of the genes in which mRNA levels correlate with MHC-I expression and tried to identify regulatory candidates for MHC-I molecules in CRC by combining this with the IFNγ signature. We hypothesized that the mRNA and protein expression levels of the regulators are associated with the expression of MHC-I molecules, and these regulators should also be involved in the IFN-γ signaling pathways. We identified a few candidate genes and confirmed their role in MHC-I expression using wet-lab experiments. We believe that targeting these regulators and increasing the expression of MHC-I molecules could potentiate the efficacy of immune therapy in CRC, thereby increasing patients’ survival chances.

## 2. Materials and Methods

### 2.1. Patient Information and Data Collection

The TCGA mRNA-sequencing data of CRC was obtained from the Broad Institute FireBrowse portal [36]. A total of 459 CRC samples were used for analysis. The patient clinical information was obtained from cBioPortal [37]. The protein expression data in cancerous and normal colon tissues was obtained from the Proteogenomic Analysis of Human Colon Cancer study [38].

### 2.2. Tumor-Infiltrated Immune Cell Fraction Analysis

The fraction of tumor-infiltrating immune cells was estimated by using quanTIseq [39], a deconvolution-based method which formulates a system of equations that describe the gene expression of a sample as the weighted sum of the expression profiles of the admixed cell types. The information can be obtained from https://icbi.i-med.ac.at/software/quantiseq/doc/, accessed on 25 September 2021. Briefly, the cell type fractions were deduced from the bulk gene expression by giving a signature matrix and using least squares regression. We used quanTIseq to analyze the normalized RNA-seq data of 459 TCGA CRC samples based on the instructions and code obtained from quanTIseq documentation (https://icbi.i-med.ac.at/software/quantiseq/doc/, accessed on 25 September 2021). The quanTIseq software directly generated the absolute fraction scores of ten cell types relative to the total cells of the tumors, including B cells, CD4 T-cells (non-regulatory CD4+ T cells), CD8+ T cells, DCs (dendritic cells), M1 Macrophages, Monocytes, Neutrophils, NK cells, Treg cells (regulatory T cells), and other uncharacterized cells.

### 2.3. Survival Analysis Based on Cell Fractions and MHC-I Molecules

Survival analysis of CRC patients stratified by the infiltrated immune cell infiltration and MHC-I molecule expression was conducted using Kaplan–Meier survival estimates. p-values were calculated using the Log-Rank test in order to determine whether there are significant differences between the two survival curves. The cell fractions of CD8+ T cells, CD4+ T cells, NK cells, the components of MHC-I molecules, mainly HLA-A, HLA-B, HLA-C, and B2M, and intracellular peptides process proteins TAP1 and TAP2 were compared for survival analysis.

### 2.4. Identifying Molecules That Might Regulate MHC-I Expression Based on Silico Analysis

Based on the TCGA mRNA-seq dataset, a list of genes that significantly correlate with *HLA-A*, *HLA-B*, *HLA-C*, *B2M*, *TAP1*, and *TAP2* mRNA levels with *p*-value < 0.05 and *r*-value > 0.3 or <−0.3 was generated and counted. Lists for MHC-I-common related genes (genes correlated with *HLA-A*, *HLA-B*, *HLA-C*, *B2M*, *TAP1*, *TAP2*) were generated first using the VennPainter software [40], used to depict sets using Venn diagrams. Then, we compared the genes in the MHC-I-TAP-common list to the IFN-γ signature genes list (obtained from Gene Set Enrichment Dataset [41]) and the final genes list. Following this, the obtained genes were compared with the data from the Proteogenomic Analysis of Human Colon Cancer list, and genes with levels that were not significantly different between normal and cancerous tissues were eliminated.

### 2.5. Gene Knockdown and Overexpression

The siRNA system was used to knockdown and the lentivirus system was used to overexpress the target genes. Briefly, siRNA-mediated knockdown of *DDX60* in NCI-H747 was generated using 4-set of siRNA smart pool (Horizon Discovery Ltd., Waterbeach, United Kingdom, Cat#, M-017664-01-0005) targeted sequences: 5′-TATAATTTGTGAGTATTCC-3, 5′-TAATAAACTGGAATATTCC-3, 5′-TATATTTATGCTAATTGAC-3, and 5′-CTTTACCACTTTCCTACGA-3′. To knockdown the gene, 3 × 10^5^ cancer cells were seeded in 6-well plate format, allowed to equilibrate overnight, and then treated with siRNA of target genes (10 µM) or nontarget control for 72 h. DDX60 overexpression was established using pLenti-GIII-CMV vector carrying DDX60 cDNA obtained from Applied Biological Materials Inc. (Richmond, BC, Canada, Cat# 178610610196). DDX60 and control Lentivirus was produced in HEK 293T cells by transfecting with DDX60 or control constructs along with psPAX2, and pMD2.G packaging constructs using Turbofect. Viral supernatants were collected 48 h post-transfection, spun for 5 min at 1200 rpm and filtered using 0.45 µm syringe filter. Target cells were infected twice by incubation in viral supernatants with 8 µg/mL polybrene. The transduced cells were selected with 2 mg/mL puromycin 48 h after infection. Protein was collected for confirming the gene overexpression and knockdown efficacy.

### 2.6. Immunoblotting

MHC-I expression and DDX60 overexpression and knockdown efficacy analysis were analyzed using Western blot. Briefly, cell lysates were prepared by scraping the cells from wells of a 6-well plate into 200 µL of RIPA lysis buffer (50 mM Tris-HCl pH 8.0 containing 1% NP-40, 150 mM NaCl, 5 mM EDTA, and 1 mM phenylmethylsulphonyl fluoride). Lysates were centrifuged at 13,000 rpm for 15 min at 4 °C. The supernatant was collected, and the total protein was estimated using Bradford’s reagent (BIO-RAD). Protein samples were separated using SDS-PAGE and transferred onto the nitrocellulose membranes. The membranes were blocked using 5% milk solution prepared in TBS-T. The membranes were incubated overnight with primary antibodies against DDX60 (Sigmaaldrich, St. Louis, MO, USA, Cat# HPA046952, 1:1000), B2M (Santa Cruz Biotechnology, Dallas, TX, USA, Cat# sc-13565, 1:2000), HLA-A (Santa Cruz Biotechnology, Inc., Dallas, TX, Cat# sc-365485, 1:3000), HLA-B/C (Santa Cruz Biotechnology, Inc., Dallas, TX, USA, Cat# sc-166668, 1:2000), and beta-actin (Cell Signaling Technology, Danvers, MA, USA, Cat# 4967, 1:5000). The blots were washed and incubated with HRP-conjugated secondary antibodies (1:5000) for 1 hr at room temperature. The chemiluminescent signal was developed with SuperSignal Pico ECL reagent (Thermo Fisher, Waltham, MA, USA).

### 2.7. Fluorescence-Activated Cell Sorting (FACS) Analysis for MHC-I Expression Following Candidate Genes’ Modification

After identifying target genes, FACS experiments were performed on tumor cells to validate the effects of these genes on MHC-I expression. Briefly, the single-cell suspensions (the control and DDX60-overexpressed stable cells, or cancer cells treated with siRNA of target genes (10 µM) or nontarget control for 72 h) were prepared by detaching the cells from cultured plates with 2 times PBS wash. The cells were collected and stained with Live/Dead Aqua stain for 10 min at room temperature. The cells were subsequently washed with PBS and stained with FITC-labeled anti-human HLA-A, B, C Antibody (Bioligend, San Diego, CA, USA, Cat# 311404) for 20 min. Next, the cells were washed with FACS buffer (5% BSA and 1 mM EDTA in PBS) and fixed with 4% paraformaldehyde. After washing, the cells were finally resuspended in FACS buffer. The stained cancer cells were then acquired and analyzed through BD FACS LSRII Y/G. The data was analyzed using FlowJo analysis software V9.

### 2.8. Statistical Analyses

The statistical analysis was calculated using GraphPad Prism software (GraphPad Software Inc., San Diego, CA, USA). For survival analysis, the survival curves were plotted using the Kaplan–Meier method; statistical differences were compared using the Mantel–Cox log-rank test. The statistical significance was determined using unpaired Student’s *t*-test (two-tailed) for two groups, one-way ANOVA test with Tukey’s multiple comparisons test for more than two groups. *p* < 0.05 was considered significant. *** denotes *p*-value < 0.001, ** denotes *p*-value < 0.01, * denotes *p*-value < 0.05.

## 3. Results

### 3.1. MHC-I Expression Is Correlated to Responders in Anti-PD1 Immunotherapy and CD8+ T Cell Infiltration in CRC

To evaluate the role of MHC-I class molecules in patients’ response to immunotherapies, we compared the expression levels of MHC-I molecules in CRC and some other solid tumor tissues from responders and non-responders to various immunotherapies, including anti-PD1, anti-CTLA4, adapt T cell transfer (ACT), and cancer vaccines [42]. As seen from Appendix A, although the relative mRNA levels of MHC-I molecules in immunotherapy responders’ tumors are variable compared with non-responders’ tumors depending on the cancer types, the gene expression levels of *HLA-A*, *HLA-B*, *HLA-C* and B2M are relatively higher in responders’ tumors than of non-responders’ tumors across most cancers, including CRC. Moreover, the expression levels of *TAP1* and *TAP2*, another two molecules essential for MHC-I-mediated antigen presentation, are also relatively higher in responders than non-responders in CRC and many other cancers (Appendix A).

MHC-I-mediated antigen presentation is essential for T cell activation. We therefore determined if the expression of MHC-I molecules correlates with CD8+ T cell infiltration in CRC. The fraction of tumor-infiltrating immune cells in 459 TCGA CRC samples was estimated by using TIMER methods based on the RNA sequencing data. As seen in Appendix A, the mRNA expression levels of MHC-I molecules (*HLA-A*, *HLA-B*, and *HLA-C*) are significantly positively correlated to the infiltration of CD8+ T cells in CRC tumors. To evaluate the role of MHC-I and immune cell infiltration in the clinical outcomes of these 459 TCGA CRC patients, we examined the effects of the infiltrated immune cells and the expression of MHC-I molecules on the patient’s prognosis in CRC. The fraction of tumor-infiltrated immune cells was analyzed by quanTIseq [39]. Survival curves of CRC patients were stratified by the higher (top 33%) or lower (bottom 33%) infiltrated immune cell infiltration. As shown in Figure 1A, CD4+ or CD8+ T cell infiltration alone did not significantly affect the survival probabilities of the CRC patients (*p* > 0.05). However, the patients have significantly better survival probability if their tumors have higher levels of infiltrated NK cells (*p* = 0.0095). Because NK cells serve to kill cells that reduce or do not display MHC-I molecules, the favorable prognostic value of NK cell infiltration suggests that the colorectal tumor cells may downregulate MHC-I to avoid T cell-mediated immune surveillance.

We then further analyzed the prognostic effect of MHC-I molecules on CRC patients. As shown in Figure 1B, the expression level of *HLA-A/B/C* does not statistically significantly affect the survival rate of CRC patients (*p* = 0.0536, nearly statistically significant, suggesting the MHC-I molecules may somewhat influence survival). However, if the colorectal tumors have both *HLA-A/B/C* expression high and T (CD4+ T and CD8+ T) cell infiltration high, the patients have a significantly better prognosis (*p* = 0.0002) (Figure 1D). Higher expression of *TAP1* and *TAP2*, another two molecules essential for MHC-I-mediated antigen presentation, is significantly correlated with better prognosis, and it is also the case when coupled with T cell expression (Figure 1E). Receiver operating characteristic (ROC) curve and area under curve (AUC) values showed that the combination of MHC-I molecules and T cell infiltration have a better predictive performance than MHC-I or T cell infiltration alone (Figure 1F,G). These results further indicate that the expression of MHC-I is required for T cell surveillance in CRC.

### 3.2. DDX60 and ISG15 Are Potential Candidates Correlated with MHC-I Expression

The comparison of the expression level of MHC-I molecules in the tumors from immunotherapy responders and non-responders suggests that MHC-I expression might be downregulated in CRC cells from non-responders (Appendix A). We then tried to screen out the potential regulators of MHC-I in CRC. Since MHC molecules are known to be coordinately expressed, we hypothesized that *HLA-A/B/C*, *B2M*, and *TAP1* and *TAP2* might have common regulators. We first checked the genes co-expressed with MHC-I molecules *HLA-A, B*, *C*, *B2M*, *TAP1* and *TAP2* based on the mRNA sequencing data of 459 TCGA CRC samples. As shown in Figure 2A, there were 96 genes that correlated with all four MHC-I genes. Using the same data and method, we found 646 genes that were associated with both TAP1 and TAP2. Among the 96 genes correlated to MHC-I genes and the 646 genes correlated to TAP genes, 86 common genes were correlated to both MHC-I and TAPs (Figure 2B,C). Then, we compared the genes in the MHC-I and TAP common list to the IFN-γ signature genes (obtained from Gene Set Enrichment Dataset) because IFN signaling is the most well-studied pathway that can stimulate the expression of MHC-I to enhance immune cell recognition. We crossed the 86 MHC-I- and TAP1 and 2-associated genes with the 500 signature genes of the IFNγ signaling pathway. A total of 57 genes were identified as being correlated with both IFNγ and MHC-I pathways (Figure 2D).

A recent proteogenomic analysis of human CRC identified more than 6000 differentially expressed molecules between tumor and normal tissues [38]. We hypothesized that MHC-I molecules are downregulated to escape immune destruction in CRC cells; the expression levels of the candidates of MHC-I regulators should be also altered in CRC compared with normal tissues. Thus, we compared the protein expression of the obtained genes using proteomics data of tumor and normal tissue to select the candidates showing more disparity. We found that there were 35 out of 57 identified common genes from the differentially expressed protein list (Figure 2E). Based on the molecular functions and further comparison of different fractions between normal tissues and cancerous tissues for disparity, we isolated two genes as potential candidates: DDX60 and ISG15 (Table 1).

### 3.3. DDX60 regulates MHC-I expression in CRC cells

To get a preview of the correlation coefficient between *ISG15*, *DDX60*, and MHC-I molecules, we generated linear regression graphs of MHC-I genes and *ISG15* or *DDX60*. The mRNA levels of *ISG15* and *DDX60* are significantly positively correlated with the expression of *B2M*, *HLA-A*, and *HLA-B* with a correlation coefficient > 0.45 (Figure 3A), confirming that *IGS15* and *DDX60* expression may influence MHC-I expression. Since the correlation analysis is based on bulk RNA-seq data, to confirm that the identified candidates DDX60 and ISG15 are expressed in CRC cells, we checked the expression level of DDX60 and ISG15 in six different CRC cells by Western-blot. We did not detect ISG15 in these cells. DDX60 showed variable levels in different cells (Figure 3B). NCI-H747 has a relatively higher DDX60 expression level than DLD-1 and HCT-115 cell lines.

To determine the role of DDX60 in regulating MHC-I, we first overexpressed DDX60 in DLD-1 and HCT-15 cells by using a lentiviral-based system. As shown in Figure 4A,B, DDX60 was successfully overexpressed in DLD-1 and HCT-15 cells. The expression levels of MHC-I molecules were upregulated in DDX60-overexpressed cells (Figure 4A,B, Appendix A). FACS analysis showed that overexpression of DDX60 increased the signal intensity of both basal level and IFNγ-induced MHC-I molecules in DLD-1 and HCT-15 cells (Figure 4C–F and Appendix A). To further confirm the role of DDX60 in regulating MHC-I, the siRNA-mediated knockdown system was used to reduce the expression of DDX60 in NCI-H747 cells. As shown in Figure 4G–I, knockdown of DDX60 reduced the protein level of MHC-I molecules and the FACS signal intensity of cell surface MHC-I in NCI-H747 cells. These results demonstrated that DDX60 can regulate the expression of MHC-I molecules.

### 3.4. DDX60 Is a Favorable Prognostic Maker and Is Downregulated in CRC

Since DDX60 can affect the expression of MHC-I molecules in CRC cells, we then checked the expression level of DDX60 in normal colon tissues and in cancerous tissues. As shown in Figure 5A, normal tissues have obviously higher expression level of DDX60 compared to CRC tissues, suggesting that DDX60 is downregulated in cancerous tissues. Furthermore, the survival analysis showed that higher expression of DDX60 is correlated with better survival probability (Figure 5B).

### 3.5. DDX60 Expression Is Correlated with Immune Cell Infiltration in CRC

The effect of DDx60 on the expression of MHC-I molecules suggests that DDX60 may influence the tumor immune microenvironment. We then examined the correlation between DDX60 expression and immune cell infiltration. The mRNA level of DDX60 is significantly positively correlated with the infiltration levels of dendritic cells (DCs), CD8+ T cells, and CD4+ T cells (Figure 6A–F). To further evaluate the role of DDX60 in immunotherapy, we compared the expression level of DDX60 in the tumors from responders and non-responders to immunotherapies. We found the responders’ s tumors have relatively higher DDX60 than tumors from non-responders (Figure 6G). From this, we conclude that DDX60 expression is correlated with immune cell infiltration and may contribute to immunotherapy in CRC.

## 4. Discussion

In recent years, immunotherapy has shown tremendous improvements in treating multiple types of cancers. There are multiple methods in immunotherapy, such as cancer vaccines, chimeric antigen receptor (CAR)-T therapy, and checkpoint inhibitors [43], all directed to improve the patient’s own immune system’s response toward cancers. Immunotherapies have induced durable responses across diverse cancers. However, more than 80% of CRC patients do not respond to current immunotherapy [44]. One of the major immune evasion mechanisms is that cancer cells lack MHC-I signals to unmask themselves, meaning that the immune cells cannot detect them. However, it is largely unknown how MHC-I molecules are downregulated in CRC [45]. Here, by combining ‘wet-lab’ experiments and analyzing that data from TCGA, GSEA, Human Protein Atlas, and CRC Proteomics and single cell-seq Datasets, our studies demonstrated that DDX60 may regulate the expression of MHC-I molecules in CRC.

Cell surface expression of MHC-peptide complexes requires properly folded MHC class I heavy chain binding to β2-m and stabilization by antigenic peptide binding in the heavy chain peptide-binding groove. Other factors involved in peptide-MHC complex assembly and intracellular transport include the molecular chaperones calnexin, calreticulin and ERp57, TAPs, and low molecular weight protein 2 (LMP2) and LMP7 [46]. Defects in this antigen processing mechanism (APM) are associated with the loss of MHC class I surface expression in cancer cells. Tumors lacking MHC class I will not be recognized by the T cells, therefore avoiding T cell-mediated immune killing. However, in the absence of surface MHC class I, these tumors are vulnerable to NK cell attack because MHC-I molecules are inhibitory ligands for KIR in NK cells [47]. Our studies showed that T cell infiltration did not significantly affect the survival probabilities of CRC patients, which is different from some previous studies. The possible reasons include different analysis models and cut-off values. We use quanTIseq and EPIC models with 33% cut-off. However, we found that NK cell levels can impact survival significantly. As cancer cells downregulate their MHC-I, they enable NK cell-mediated killing. Considering the survival-promoting role of NK cell infiltration in CRC, NK cell-based immunotherapy may provide an alternative approach for CRC treatment.

DDX60 is a DEAD box protein, which plays important roles in fundamental RNA-mediated processes by using hydrolysis [48,49]. It has been confirmed in other studies that DDX60 is correlated with IFN-γ [50], a cytokine secreted by T cells whose functions consist of enhancing antigen presentation and coordinating the innate immune system, among other things. It has also been demonstrated that DDX60 is involved in the activation of the RIG-I pathway [51], whose receptors play a role in sensing virus infection and mediating some gene transcriptions. Previous studies showed that treatment with RIG-I agonist resulted in the activation of proinflammatory transcription factors STAT1 and NF-κB and increased expression of MHC-I components in cancer cells [52,53], Therefore, DDX60 may serve as a signaling transductor to activate the RIG-I pathway. In addition, DDX60 might be able to sense the abnormal cytosolic RNA (viral, bacterial, or self-RNA) and activate MHC-I signaling in cancer cells [54,55]. Our studies showed that manipulation of DDX60 expression could significantly affect the expression of MHC-I class molecules. The expression of *DDX60* is a favorable biomarker for the survival of CRC patients. Moreover, CRC tumors with high expression of DDX60 show more proinflammatory phenotypes (e.g., increased infiltration of DCs, CD4+, and CD8+ T cells) compared with low DDX60 tumors. These results indicated that DDX60 might have important clinical significance in CRC and could serve as a potential immune therapeutic target [56,57]. However, it is relevant to note that the exact molecular mechanism(s) by which DDX60 may regulate MHC-I is still unclear. Existing studies have shown that the expression of MHC-I genes can be regulated through transcription factors, epigenetic modifications, and post-transcriptional and post-translational mechanisms [14]. Further research is needed on whether DDX60 affects MHC-I via these pathways.

The major limitations of our study are that (1) our data does not answer the question of how DDX60 regulates MHC-I molecules, (2) we do not have direct evidence to support whether targeting DDX60 can enhance CD8+ T cell-mediated immunotherapies, and (3) the analyzed patient data could have come from CRC patients with different backgrounds (e.g., sex, age, baseline diseases, and pre-treatments), limiting the accuracy of our analysis. In the future, more work could be focused on the molecular mechanism of how DDX60 regulates MHC-I signaling and the effect of DDX60 on immunotherapies.

In summary, we analyzed the effect of MHC-I molecules on survival probabilities in CRC. Through analysis of MHC-I co-expressed genes, we identified several candidates for further “wet-lab” confirmation. Our results demonstrated that the expression of MHC-I molecules is associated with patient prognosis; DDX60 regulates the expression of MHC-I molecules; DDX60 expression is down-regulated in CRC and is a favorable prognostic marker. Thus, targeting DDX60 may improve the effects of immunotherapy on some patients.

## Figures and Tables

**Figure 1 biomedicines-10-03092-f001:**
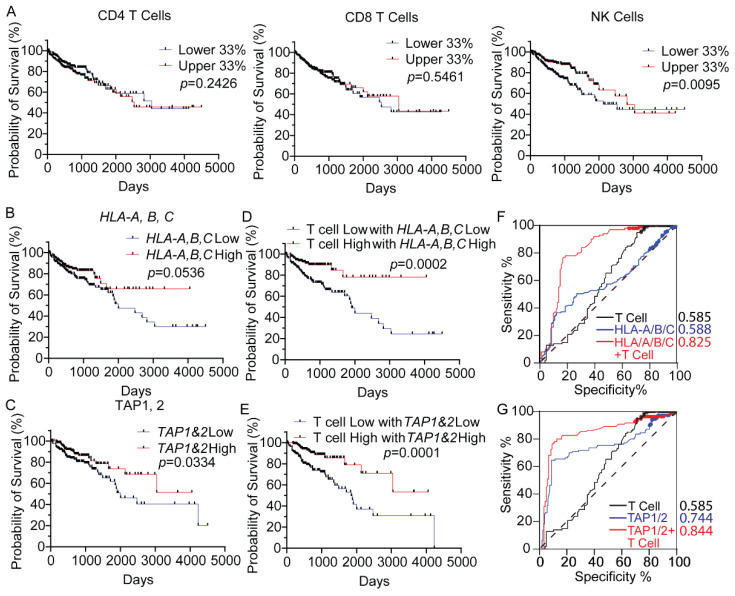
**Expression of MHC-I molecules is associated with patient prognosis in CRC**. (**A**) Kaplan–Meier curves showing the survival probability of 459 TCGA CRC patients based on the tumor infiltrating CD4 T cell, CD8 T cell, and NK cell levels. Survival data are from samples with higher (upper 33%) and lower (lower 33%) immune cell fraction. (**B**) Kaplan–Meier curves showing the survival probability of CRC patients based on *HLA-A*, *B*, *C* expression levels. (**C**) Kaplan–Meier curves showing the survival probability of CRC patients based on *HLA-A*, *B*, *C* expression levels and T cell levels. (**D**) Kaplan–Meier curves showing the survival probability of CRC patients based on *TAP1*, *2* expression levels. (**E**) Kaplan–Meier curves showing the survival probability of CRC patients based on *TAP1*, *2* expression levels in patients with low T cell levels. (**F**,**G**) Receiver operating characteristic (ROC) curve and area under curve (AUC) values for the prognostic prediction model based on T cell infiltration and MHC-I expression.

**Figure 2 biomedicines-10-03092-f002:**
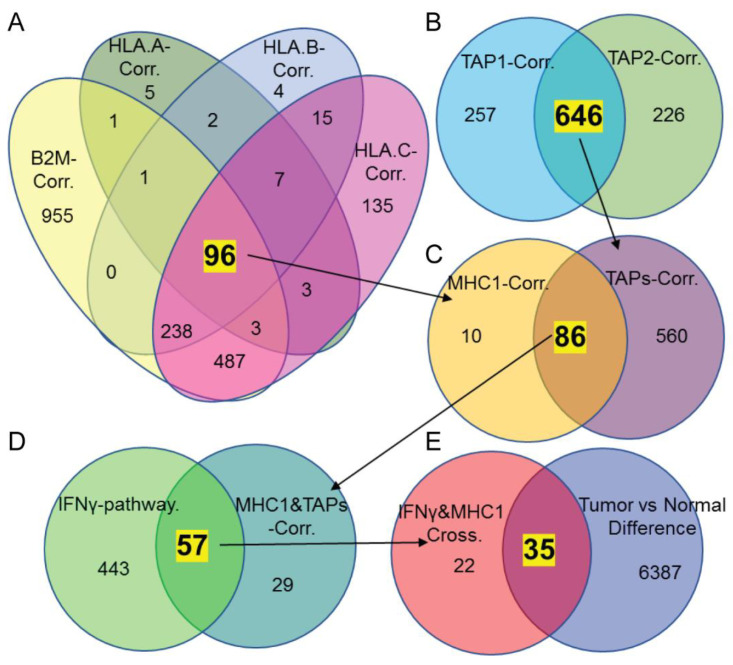
**Potential candidates correlated with MHC-I expression.** (**A**) Venn diagrams showing the commonly correlated genes for *HLA-A*, *B*, *C*, and *B2M* based on TCGA mRNA sequencing data of CRC (*r* > 0.3, or ≤0.3, *p* < 0.05). (**B**) Venn diagrams showing the commonly correlated genes for *TAP1* and *TAP2*. (**C**) Venn diagrams showing the common correlated genes from (**A**,**B**). (**D**) Venn diagrams showing the common correlated genes analysis for MHC-I-related molecules and IFN-γ pathways. (**E**) Venn diagrams showing the common genes from (**D**) and differentially expressed molecules identified by the Proteogenomic Analysis of Human Colon Cancer study obtained from the study published in [38].

**Figure 3 biomedicines-10-03092-f003:**
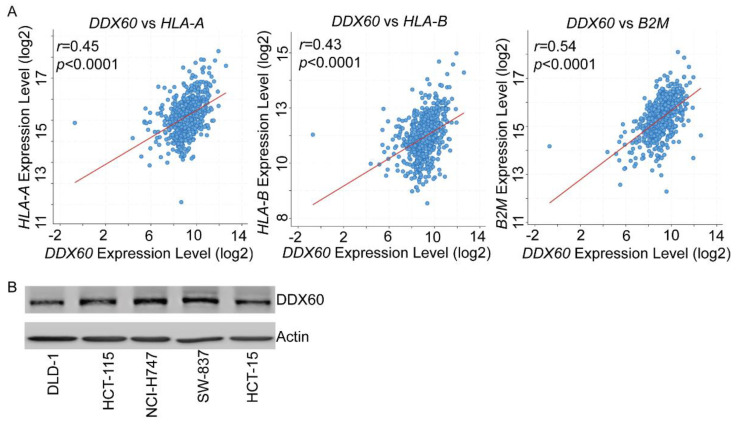
**The expression of *DDX60* is correlated with MHC-I expression in CRC.** (**A**) Linear regression analysis showing the correlation between *DDX60* expression levels (log2) and MHC-I-correlated molecules expression levels (log2) based on mRNA sequencing data of 459 TCGA CRC samples. (**B**) Western blotting images showed the protein levels of DDX60 and Actin in six different CRC cell lines.

**Figure 4 biomedicines-10-03092-f004:**
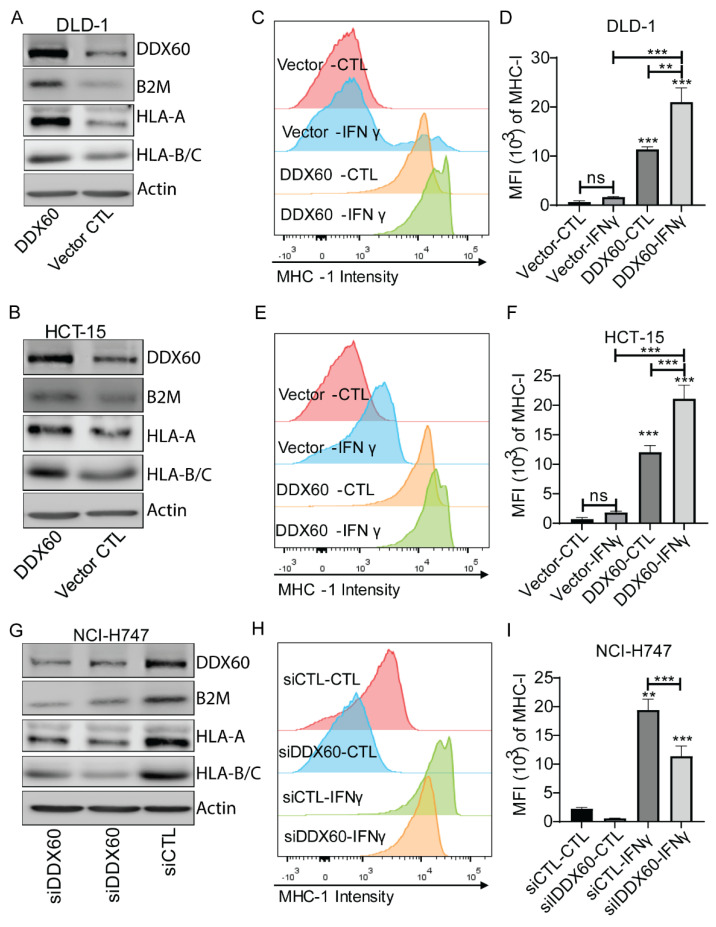
**DDX60 regulates the expression of MHC-I molecules in CRC cells.** (**A**,**B**). Western blotting images showed the expression of DDX60, B2M, and Actin in vector control or DDX60-overexpressed DLD-1 (**A**) and HCT-15 (**B**). (**C**–**F**) FACS analysis of MHC-I expression in vector control or DDX60-overexpressed DLD-1 (**A**) and HCT-15 (**B**) with/without IFNγ treatment. (**G**) Western blotting images showed the expression of DDX60, B2M, and Actin in NCI-H747 cells treated with siRNA of nontarget control (siCTL) or *DDX60* for 48 h. (**H**,**I**) FACS analysis of MHC-I expression in control (siCTL) and DDX60-Knockdown NCI-H747 cells with/without IFNγ stimulation. *p* values were calculated by one-way analysis of variance (ANOVA) with Tukey’s multiple comparison. ns: *p* ≥ 0.05; **: *p* < 0.01; ***: *p* < 0.001.

**Figure 5 biomedicines-10-03092-f005:**
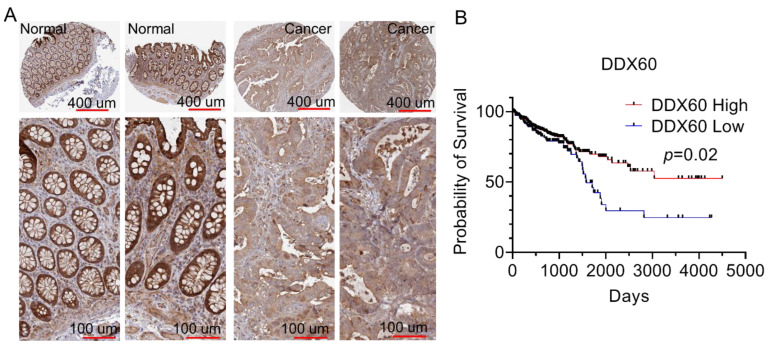
**DDX60 is downregulated in CRC and its expression is associated with patient prognosis.** (**A**,**B**). Immunohistochemical staining images showing the expression level of DDX60 in normal and cancerous tissue. Data were obtained from Protein Atlas (https://www.proteinatlas.org/, accessed on 14 January 2022) ENSG00000137628-DDX60/pathology/colorectal+cancer. (**B**) Kaplan–Meier survival curves showing the survival probability of CRC patients with high or low DDX60 levels. Data were obtained from TGCA.

**Figure 6 biomedicines-10-03092-f006:**
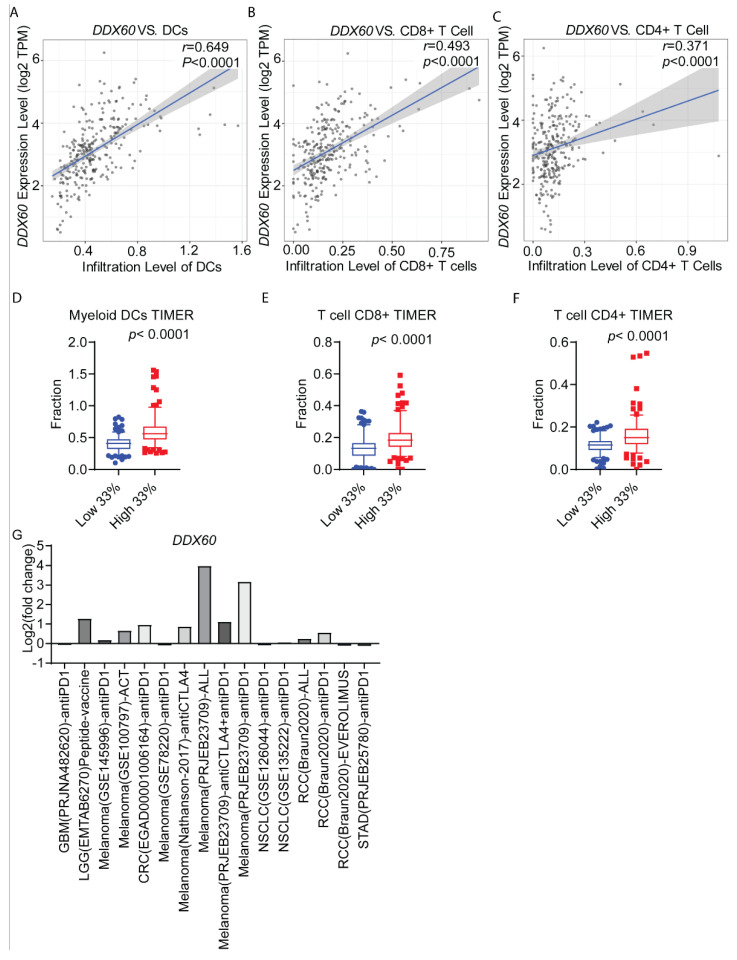
**DDX60 expression is correlated with immune cell infiltration and immunotherapy response.** (**A**–**C**) The correlation analysis of DDX60 expression and immune cell infiltration of dendritic cells (DCs), CD8+ T cells, and CD4+ T cells based on TCGA CRC RNA-seq data. (**D**–**F**) Immune cell infiltration of dendritic cells (DCs), CD8+ T cells, and CD4+ T cells in CRC tissues with high or low *DDX60* expression. (**G**) The mRNA expression levels (fold change of Log2) of *DDX60* in the tumors from the immunotherapy responders and non-responders in different solid cancer. The gene expression data were obtained from Tumor Immunotherapy Gene Expression Resource (TIGER) at http://tiger.canceromics.org/#/home accessed on 10 May 2022.

**Table 1 biomedicines-10-03092-t001:** The different fractions between normal tissues and cancerous tissues (from the Proteogenomic Analysis of Human Colon Cancer Study [38] for the 35 potential candidates correlated with MHC-I and IFN-γ expression. The genes highlighted in yellow are the final candidates.

Gene Name	High Expression in Normal		Cases of High Expression in Cancer	
	Cases	Fraction (%)	Cases	Fraction (%)
GZMA	77	79.38	20	20.62
HLA-DPA1	76	78.35	21	21.65
APOL1	74	76.29	23	23.71
DDX60	73	75.26	24	24.74
ISG15	68	70.10	29	29.90
BTN3A3	64	65.98	33	34.02
CMPK2	64	65.98	33	34.02
HLA-DMB	61	62.89	35	37.11
ISG20	61	62.89	36	37.11
DDX58	57	58.76	40	41.24
APOL3	55	56.70	42	43.30
HLA-DRA	55	56.70	42	43.30
HLA-DRB1	54	55.67	43	44.33
HLA-F	51	52.58	46	47.42
IFI35	49	50.52	48	49.48
TRIM22	49	50.52	48	49.48
PARP12	48	49.48	49	50.52
IFIT3	46	47.42	51	52.58
OASL	42	43.30	55	56.70
APOL2	37	38.14	60	61.86
CD74	36	37.11	61	62.89
GBP4	34	35.05	63	64.95
HLA-E	34	35.05	63	64.95
GBP2	34	35.05	63	64.95
GBP1	33	34.02	64	65.98
SERPINB9	33	34.02	64	65.98
PARP14	30	30.93	67	69.07
PARP9	27	27.84	70	72.16
UBE2L6	27	27.84	70	72.16
PSMB9	25	25.77	72	74.23
TYMP	21	21.65	76	78.35
STAT1	15	15.46	82	84.54

## Data Availability

The data presented in this study are openly available in TCGA database. The raw data, processed data, and clinical data can be found in the legacy archive of the National Cancer Institute’s Genomic Data Commons (GDC) (https://portal.gdc.cancer.gov/legacy-archive/search/f accessed on 20 September 2021).

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
