# Peer review of "Identification of DDX60 as a Regulator of MHC-I Class Molecules in Colorectal Cancer"

_biomedicines, 2022, doi:10.3390/biomedicines10123092_

Round 1
Reviewer 1 Report
1. In this study, authors indicated that overexpression of DDX60 upregulated MHC-I expression while knockdown of DDX60 reduced the MHC-I expression in CRC cells. In the case of KO DDX60, authors might be measured the cytosolic metabolites in this cell lines.
2. Thei study suggested that DDX60 could regulate MHC-I expression in CRC; thus, targeting DDX60 may improve the effects of immunotherapy on some patients. Do you have any drugs to regulate the DDX60 expression?
Author Response
Comments and Suggestions for Authors
- 1. In this study, authors indicated that overexpression of DDX60 upregulated MHC-I expression while knockdown of DDX60 reduced the MHC-I expression in CRC cells. In the case of KO DDX60, authors might be measured the cytosolic metabolites in this cell lines.
Author's Response: We thank the reviewer’s suggestion. This is a very interesting direction for studying the role of DDX60 in cancer cells. The activation of DDX60 has been shown to inhibit viral RNA replication (Oliver Grünvogel, et. al., 2015) and reduce translation of viral ribosome (Mohammad Sadic, et. al., 2022), indicating that DDX60 may regulate the metabolism of amino acids and nucleotides. We have some preliminary data showing that the activation of DDX60 induces the downregulation of mTORC1, a key regulator of intracellular metabolic activity. We didn’t discuss these findings here because the present study was simply focused on the effect of DDX60 on levels of MHC-I molecules. One of our current projects is to determine if and how mTOR signaling is involved in DDX60on regulating MHC-I molecules.
- 2. The study suggested that DDX60 could regulate MHC-I expression in CRC; thus, targeting DDX60 may improve the effects of immunotherapy on some patients. Do you have any drugs to regulate the DDX60 expression?
Author's Response: Grate question but there is not a compound that can specifically target the DDX60. We are currently employing a Drugs Library screening approach with FACS sorting to identify factors that increases both DDX60 and MHC-I. The preliminary data from the screening suggests that some drugs like Birinapant, Azacitidine, and Ciclopirox can stimulate DDX60 and MHC-I expression in colorectal cancer cells, but the drug action mechanisms are largely unknown. We are trying to figure out the common mechanisms behind the inductive effect.
Reviewer 2 Report
The manuscript authored by Geng et al., addresses the main mechanisms which results in down regulation of MHC-I epitope presentation to CD8+ T cells, thus resulting in loss of patient response to immunotherapy. The biological question is of significance as, indeed restoring the ability for MHC-I peptide presentation could increase the immune response in CRC. However, they start by analysis of patient data to show that MHC-I is downregulated in non-responders and that its expression is associated with CD8+ T cell infiltration. Although not convincing (I have detailed below my concerns regarding their data) these events are somewhat expected, and it is not something of significant novelty to the field so in my opinion these could go in the supplemental. Moreover DDX60 and ISG15 have been previously linked to MHC-I expression (see my concerns) so this is also not new. The rest of the manuscript contains data from publically available databases. Although the main goal of the authors is of interest, they do not bring sufficient novelty to warrant publication. Moreover they should really double-check their manuscript text and figures as I find numerous mistakes (for example Figure 9G displays log2 FC for DDX60 but in the figure legend is mentioned MHC-I). Although their strategy would seem to be logic, performing first an in silico screening and then validating in wet-lab experiments their findings and conclusions are mostly based on querying various public sources and comparing lots of things (bulk proteomics with single cell mRNA data). The authors should also focus first on detailing about how do they exactly select their targets for their wet-lab experiments (their exactly criteria). This is not convincingly described in the manuscript. Moreover there are also some errors in their analysis so they should thoroughly readdress this problem in my opinion. Considering these observations and my concerns detailed below I cannot recommend publication. However, the main goal is of interest so probably a resubmission could be considered, providing more wet-lab mechanistic results that would up-scale the manuscript novelty.
Major points:
1. The authors describe briefly in the Introduction section the molecular pathways, which results in alteration of the MHC-I expression level, emphasizing the link between IFN and MHC-I level. However they should at least mention that the MHC-I level can also be regulated post-translational, at the protein level, by retention in endoplasmic reticulum and further degradation as was previously shown (see PMIDs: 23929775, 34332121). Thus restoring MHC-I level would not in all cases result in a better immune response, as the target proteins would be retained intracellular and sent to degradation. This is an important aspect, which should be addressed by the authors.
2. Based on the results from Figure 1 the authors claim that MHC-I expression level is correlated to responder in immunotherapy. However they start referring in the introduction that CRC patients do not respond to immunotherapy but show this data on different cancers (lung cancer, glioblastoma etc.). They should emphasize these for CRC patients. Moreover, the analyzed data could have come from patients with different backgrounds. For example it is currently known that the immune response is also age-dependent (see PMIDs: 34433020, 30906662). Also additional chemotherapeutic/surgical interventions/radiotherapy etc could bias their results. They should reorganize and re-analyze their data to draw such conclusions and maybe limit only to CRC as this is the focus of their manuscript.
3. Figure 2: The authors should try to explain their findings from the data in Figure 2. They should give more details, as to which malignant cells were analyzed: human or mouse (the referenced studied mentioned both), what does TPM stand for? Why is log(TPM/10+1)? What are x and y axis? Although I do agree UMAP plots are frequently used, in the current context the authors do not provide sufficient information. Just providing a figure without explaining in the text and/or figure legend their message does not bring any value to their manuscript. Alternatively, they could skip this data or merge it with Figure 1.
4. Figure3: Again, which patients were used for CD8+ T cell assessment and HLA-A, B and C? Are we comparing the same patient MHC-I expression and CD8+ T cells? Are these only TILs or circulating? The correlation is limited and clearly there are also other factors which limit T cell infiltration and MHC-I expression level. As mentioned before MHC-I degradation level (it is currently known that there is limited correlation between mRNA and protein expression level), but also tumor vasculature and microenvironment (for ex. see PMID: 26216636). Clearly these should be at least indicated.
5. What is the exact logic of comparing bulk proteomics data of tumor vs. normal tissue with single cell-level mRNA data for IFNg, MHC-I and TAP data genes? The authors should also explain better in their manuscript as how exactly they found DDX60 (an inducer involved in type I interferon expression) and ISG15 as potential hits and their motivation to choose these particular targets for their wet-lab experiments. These are not sufficiently addressed in the manuscript. Moreover DDX60 and ISG15 have been previously reported as associated with MHC-I expression (see PMID: 34178649, 25749047,32686110) so this is not something new.
Minor points:
1. Introduction section: There are some spelling mistakes when citing references (e.g see L46, 50, 55 etc).
2. I would advise authors to re-check their text as for some parts there is no logic (e.g. p6, L212-214: “MHC-I-mediated antigen presentation is essential for CD8+ T cell activation as well as T cell-mediated cancer cell elimination, we then determined the…”). Also see p7, L261-262, p8, L277-279, p11, L345-347.
3. Figure 4 and the corresponding text: The authors mention “But if the colorectal tumors have both HLA-A/B/C expression high and T (CD4+ T and CD8+ T) cell infiltration high, the patients have a significantly better prognosis (p = 0.0002) (Figure 4C).”, however p=0.0002 is shown in Figure 4D.
4. Figure 5: There seems to be an error in the analysis of the author’s data. They mention that 96 genes were found correlated at the mRNA expression level to MHC-I genes and 646 to both of TAP. However judging by their third Venn Diagram (C) they have 86+18=104 and not 96 genes as described in (A) for their MCH-I genes. So clearly they should double-check their analysis.
5. Figure 6A: Is this single cell mRNA-seq? Please provide details about how this analysis was performed and from where this data is (cancer cell lines, tissue, biological fluids, patients etc).
6. Figure 6B: Please also provide full Western Blots (WBs) (as supplementary) and add the molecular weight marker.
7. Figure 7A&B: How many WB experiments were performed? To claim that B2M is up-regulated please add quantification panels, statistical analysis and clearly state how many experiments were performed. For FACS data please add representative plots with stained cell populations.
8. Figure 9G: The title of the figure suggests DDX60 mRNA expression level was analyzed but the figure legend mentions MHC-I. Please correct.
Author Response
The manuscript authored by Geng et al., addresses the main mechanisms which results in down regulation of MHC-I epitope presentation to CD8+ T cells, thus resulting in loss of patient response to immunotherapy. The biological question is of significance as, indeed restoring the ability for MHC-I peptide presentation could increase the immune response in CRC. However, they start by analysis of patient data to show that MHC-I is downregulated in non-responders and that its expression is associated with CD8+ T cell infiltration. Although not convincing (I have detailed below my concerns regarding their data) these events are somewhat expected, and it is not something of significant novelty to the field so in my opinion these could go in the supplemental. Moreover DDX60 and ISG15 have been previously linked to MHC-I expression (see my concerns) so this is also not new. The rest of the manuscript contains data from publically available databases. Although the main goal of the authors is of interest, they do not bring sufficient novelty to warrant publication. Moreover they should really double-check their manuscript text and figures as I find numerous mistakes (for example Figure 9G displays log2 FC for DDX60 but in the figure legend is mentioned MHC-I). Although their strategy would seem to be logic, performing first an in silico screening and then validating in wet-lab experiments their findings and conclusions are mostly based on querying various public sources and comparing lots of things (bulk proteomics with single cell mRNA data). The authors should also focus first on detailing about how do they exactly select their targets for their wet-lab experiments (their exactly criteria). This is not convincingly described in the manuscript. Moreover there are also some errors in their analysis so they should thoroughly readdress this problem in my opinion. Considering these observations and my concerns detailed below I cannot recommend publication. However, the main goal is of interest so probably a resubmission could be considered, providing more wet-lab mechanistic results that would up-scale the manuscript novelty.
Author's Response: We thank reviewer for her/his comments that “the main goal is of interest”. We also appreciate her/his constructive comments on how to improve our work. We reorganized the data and added more details in the introduction, method, and discussion parts. We also moved the several figures into supplementary figures. Below is the point-to-point response to reviewer’s concerns.
Major points:
- 1. The authors describe briefly in the Introduction section the molecular pathways, which results in alteration of the MHC-I expression level, emphasizing the link between IFN and MHC-I level. However, they should at least mention that the MHC-I level can also be regulated post-translational, at the protein level, by retention in endoplasmic reticulum and further degradation as was previously shown (see PMIDs: 23929775, 34332121). Thus, restoring MHC-I level would not in all cases result in a better immune response, as the target proteins would be retained intracellular and sent to degradation. This is an important aspect, which should be addressed by the authors.
Author's Response: We thank the reviewer for pointing out the lack of enough discussion about the MHC-I regulation at different levels in our manuscript. We have added more information in the discussion part regarding this topic and cited the relative papers. Besides, following the reviewers’ suggestion, we also checked the expression of MHC-1 at the transcriptional level (by RT-PCR), total protein level (by Western blot), and surface protein level (by FACS). These result has been added to the manuscript.
- 2. Based on the results from Figure 1 the authors claim that MHC-I expression level is correlated to responder in immunotherapy. However, they start referring in the introduction that CRC patients do not respond to immunotherapy but show this data on different cancers (lung cancer, glioblastoma etc.). They should emphasize these for CRC patients. Moreover, the analyzed data could have come from patients with different backgrounds. For example it is currently known that the immune response is also age-dependent (see PMIDs: 34433020, 30906662). Also additional chemotherapeutic/surgical interventions/radiotherapy etc could bias their results. They should reorganize and re-analyze their data to draw such conclusions and maybe limit only to CRC as this is the focus of their manuscript.
Author's Response: We thank the reviewer for point the limitations of the analysis of Figure 1. We were trying to re-analyze the data following the reviewer’s suggestion. Unfortunately, most of these immune therapy resistance studies do not have enough patient samples (or detailed information) for conducting the subgroup re-analysis. We moved this figure to a supplementary file as indirective support data. We included CRC and another 15 studies to show the effect of MHC-I molecules on immunotherapy response is different depending on the cancer type and MHC-I molecules are higher in response individuals than non-response patients. We discussed these points in the manuscript. We also discussed the limitation of these data in the discussion part.
- Figure 2: The authors should try to explain their findings from the data in Figure 2. They should give more details, as to which malignant cells were analyzed: human or mouse (the referenced studied mentioned both), what does TPM stand for? Why is log (TPM/10+1)? What are x and y axis? Although I do agree UMAP plots are frequently used, in the current context the authors do not provide sufficient information. Just providing a figure without explaining in the text and/or figure legend their message does not bring any value to their manuscript. Alternatively, they could skip this data or merge it with Figure 1.
Author's Response: We thank the reviewer for pointing out these questions. This figure shows the expression of MHC-I in different cell components of CRC. We agree that these data just added limited evidence to the conclusion. We removed this data from the manuscript.
- 4. Figure3: Again, which patients were used for CD8+ T cell assessment and HLA-A, B and C? Are we comparing the same patient MHC-I expression and CD8+ T cells? Are these only TILs or circulating? The correlation is limited and clearly there are also other factors which limit T cell infiltration and MHC-I expression level. As mentioned before MHC-I degradation level (it is currently known that there is limited correlation between mRNA and protein expression level), but also tumor vasculature and microenvironment (for ex. see PMID: 26216636). Clearly these should be at least indicated.
Author's Response: We compared the correlation of mRNA level of MHC-I molecules and CD8+ T cell infiltration in CRC from TCGA colorectal cancer patients. We are indeed comparing the same patients for MHC-I expression and CD8+ T cells. The figures concern TILs. This information has been added in the figure description and we discussed the limitations in the discussion part.
- 5. What is the exact logic of comparing bulk proteomics data of tumor vs. normal tissue with single cell-level mRNA data for IFNg, MHC-I and TAP data genes? The authors should also explain better in their manuscript as how exactly they found DDX60 (an inducer involved in type I interferon expression) and ISG15 as potential hits and their motivation to choose these particular targets for their wet-lab experiments. These are not sufficiently addressed in the manuscript. Moreover DDX60 and ISG15 have been previously reported as associated with MHC-I expression (see PMID: 34178649, 25749047,32686110) so this is not something new.
Author's Response: Since MHC molecules are known to be coordinately expressed, we hypothesized that they might have common regulators. We first checked the genes co-expressed with MHC-I molecules HLA-A, B, C, B2M, TAP1, and TAP2 based on the TCGA mRNA sequencing data of CRC. Then, we compared the genes in the MHC-I and TAP-common list to the IFN-γ signature genes (obtained from Gene Set Enrichment Dataset) because IFN signaling is the most well-studied pathway that can stimulate the expression of MHC-I to enhance immune cell recognition. Since we hypothesized that MHC-I molecules are downregulated to escape immune destruction in CRC compared with normal tissues, the expression of the “regulatory candidates” is also possibly altered in CRC. So, we compared the protein expression of the obtained genes using proteomics data of tumor and normal tissue to select the candidates showing more disparity.
DDX60 and ISG15 are ranked at the top of the comparison list. Previous studies suggest that both of them are involved in the regulation of the immune response, such as signal transducer and activator of transcription 1 (STAT1), JAK1, and RIG-I. Loss of DDX60 and ISG15 may result in immune evasion of some viruses. Combined these results and evidence. We chose DDX60 and ISG15 to conduct the further wet-lab tests. We discussed and cited these studies indicating DDX60 and ISG15 might be associated with MHC-I expression, but our study aims to target MHC-I in CRC immunotherapy specifically; therefore, the information concerning DDX60 is accordingly novel.
Minor points:
- Introduction section: There are some spelling mistakes when citing references (e.g see L46, 50, 55 etc).
Author's Response: We thank the reviewer for pointing this out. They have been fixed.
- I would advise authors to re-check their text as for some parts there is no logic (e.g. p6, L212-214: “MHC-I-mediated antigen presentation is essential for CD8+ T cell activation as well as T cell-mediated cancer cell elimination, we then determined the…”). Also see p7, L261-262, p8, L277-279, p11, L345-347.
Author's Response: We have revised the mentioned parts and we have made corrections accordingly.
- Figure 4 and the corresponding text: The authors mention “But if the colorectal tumors have both HLA-A/B/C expression high and T (CD4+ T and CD8+ T) cell infiltration high, the patients have a significantly better prognosis (p = 0.0002) (Figure 4C).”, however p=0.0002 is shown in Figure 4D.
Author's Response: We thank the reviewer for pointing this out. The correction has been made.
- Figure 5: There seems to be an error in the analysis of the author’s data. They mention that 96 genes were found correlated at the mRNA expression level to MHC-I genes and 646 to both of TAP. However, judging by their third Venn Diagram (C) they have 86+18=104 and not 96 genes as described in (A) for their MCH-I genes. So clearly they should double-check their analysis.
Author's Response: We thank the reviewer for pointing this out. The correction has been made.
- Figure 6A: Is this single cell mRNA-seq? Please provide details about how this analysis was performed and from where this data is (cancer cell lines, tissue, biological fluids, patients etc).
Author's Response: Figure 6A is not single-cell mRNA-seq, it was obtained from TCGA datasets. We have added the necessary details in the figure legend.
- Figure 6B: Please also provide full Western Blots (WBs) (as supplementary) and add the molecular weight marker.
Author's Response: Providing full Western Blots is a mandatory requirement by the Journal. The Western Blot figures have been added in the supplementary file.
- Figure 7A&B: How many WB experiments were performed? To claim that B2M is up-regulated please add quantification panels, statistical analysis and clearly state how many experiments were performed. For FACS data please add representative plots with stained cell populations.
Author's Response: We performed all the WB and FACS experiments more than four times. We have added the necessary information and results according to the review’s suggestions.
- Figure 9G: The title of the figure suggests DDX60 mRNA expression level was analyzed but the figure legend mentions MHC-I. Please correct.
Author's Response: The correction has been made.
Reviewer 3 Report
Main point
Is the effect of DDX60 on MHC I expression at the transcriptional level or translational level? or is the effect post translational impacting cell surface presentation of HLA complex? Can the authors include qPCR data and western blot for HLA genes in the DDX60 overexpression and knockdown system? This should provide a hint at which level DDX60 operates. This data would further add to mechanistic understanding of how DDX60 regulates MHC I expression.
Minor
Figure 7G; Please correct DDX60 blot labelling
Author Response
Comments and Suggestions for Authors
Main point
Is the effect of DDX60 on MHC I expression at the transcriptional level or translational level? or is the effect post translational impacting cell surface presentation of HLA complex? Can the authors include qPCR data and western blot for HLA genes in the DDX60 overexpression and knockdown system? This should provide a hint at which level DDX60 operates. This data would further add to mechanistic understanding of how DDX60 regulates MHC I expression.
Author's Response: We thank the reviewer for pointing out the lack of enough discussion about the MHC-I regulation at different levels in our manuscript. We have added more information in the discussion part regarding this topic and cited the relative papers. Besides, following the reviewers’ suggestion, we also checked the expression of MHC-1 at the transcriptional level (by RT-PCR), total protein level (by Western blot), and surface protein level (by FACS). These results have been added to the manuscript.
Minor
Figure 7G; Please correct DDX60 blot labelling
Author's Response: The correction has been made.
Round 2
Reviewer 2 Report
Geng et al., provided a revised version of their manuscript. Although their manuscript is of limited novelty as ISG15 has been previously linked to immune escape in CRC immunotherapy (see the references from my previous report) and even more DDX60 was also found previously among other proteins to regulate CRC immunological response (see PMID 33627378) the authors addressed most of my concerns. There are some minor aspects that I have detailed below:
Major:
1. I cannot find the Western Blot quantification panels and statistical analysis mentioned by the authors. These should be added in the main figure or in the supplementary section.
Minor:
1. Also the original Western Blots with the molecular weight annotation are not in the supplementary section.
2. p3, l91-93: Ubiquitinated MHC-I molecules are not directly subject to degradation in the endoplasmic reticulum as the authors mention. These are retrotranslocated and subject to proteasomal degradation, as shown in ref 35. Please correct accordingly.
3. Figure 4 legend: The authors mention "(I)&(J) FACS analysis of ...", but the supplied figure is only A-I. Please double-check and correct accordingly.
Author Response
Reviewer 2
Geng et al., provided a revised version of their manuscript. Although their manuscript is of limited novelty as ISG15 has been previously linked to immune escape in CRC immunotherapy (see the references from my previous report) and even more DDX60 was also found previously among other proteins to regulate CRC immunological response (see PMID 33627378) the authors addressed most of my concerns. There are some minor aspects that I have detailed below:
Authors’ response: We thank Reviewer #2 for the constructive suggestions to improve our original manuscript and the positive comments on our revised version.
Major:
- I cannot find the Western Blot quantification panels and statistical analysis mentioned by the authors. These should be added in the main figure or in the supplementary section.
Authors’ response: We thank the reviewer for mentioning this issue. We added these data with the western blot supplement original data, which is separate file. We moved the quantification data to the supplementary file.
Minor:
- Also the original Western Blots with the molecular weight annotation are not in the supplementary section.
Authors’ response: We thank the reviewer for their observation. As we mentioned, these data were added in the file of western blot supplement original data, which is separate file.
- p3, l91-93: Ubiquitinated MHC-I molecules are not directly subject to degradation in the endoplasmic reticulum as the authors mention. These are retrotranslocated and subject to proteasomal degradation, as shown in ref 35. Please correct accordingly.
Authors’ response: We thank the reviewer for pointing out this issue. The necessary modifications have been brought.
- Figure 4 legend: The authors mention "(I)&(J) FACS analysis of ...", but the supplied figure is only A-I. Please double-check and correct accordingly.
Authors’ response: We thank the reviewer for pointing that out. The issue has been fixed.
Reviewer 3 Report
The authors have sufficiently addressed the concerns raised previously.
Author Response
Reviewer 3
The authors have sufficiently addressed the concerns raised previously.
Authors’ response: We thank Reviewer #2 for the constructive suggestions to improve our original manuscript and the positive comments on our revised version.